# Endosomal mTORC2 Is Required for Phosphoinositide-Dependent AKT Activation in Platelet-Derived Growth Factor-Stimulated Glioma Cells

**DOI:** 10.3390/cancers13102405

**Published:** 2021-05-16

**Authors:** Suree Kim, Sukyeong Heo, Joseph Brzostowski, Dongmin Kang

**Affiliations:** 1Fluorescence Core Imaging Center, Department of Life Science, Ewha Womans University, 52 Ewhayeodae-gil, Seodaemun-gu, Seoul 03760, Korea; kimsuree@hanmail.net (S.K.); hcjm123@hanmail.net (S.H.); 2Twinbrook Imaging Facility, National Institutes of Health/NIAID, Bethesda, MD 20892, USA; brzostowskij@niaid.nih.gov

**Keywords:** AKT, endocytosis, endosome, mTORC2, phosphatidylinositol

## Abstract

**Simple Summary:**

The full activation of AKT, which is necessary for cell physiological changes, is achieved through the phosphorylation of Thr308 and Ser473 in human AKT. Here, we have addressed how AKT activation at early endosomes occurs during growth factor stimulation and how mTORC2 is recruited into endosomes and associated with AKT. The explanation comes from the discovery of three important events: (1) the physical association of mSIN and Rictor, critical components for mTORC2 assembly and activity, with early endosomes; (2) the control of the recruitment of mSIN to endosomes by PtdIns(3,4)P_2_; and (3) the PtdIns(3,4)P_2_-mediated endosomal AKT activation through phosphorylation at Ser473 to control a subset of AKT substrates.

**Abstract:**

The serine/threonine kinase AKT is a major effector during phosphatidylinositol 3-kinase (PI3K)-driven cell signal transduction in response to extracellular stimuli. AKT activation mechanisms have been extensively studied; however, the mechanism underlying target of rapamycin complex 2 (mTORC2) phosphorylation of AKT at Ser473 in the cellular endomembrane system remains to be elucidated. Here, we demonstrate that endocytosis is required for AKT activation through phosphorylation at Ser473 via mTORC2 using platelet-derived growth factor-stimulated U87MG glioma cells. mTORC2 components are localized to early endosomes during growth factor activation, and the association of mTORC2 with early endosomes is responsible for the local activation of AKT, which is critical for specific signal transduction through glycogen synthase kinase-3 beta and forkhead box O1/O3 phosphorylation. Furthermore, endosomal phosphoinositide, represented by PtdIns(3,4)P_2_, provides a binding platform for mTORC2 to phosphorylate AKT Ser473 in endosomes through mammalian Sty1/Spc1-interacting protein (mSIN), a pleckstrin homology domain-containing protein, and is dispensable for AKT phosphorylation at Thr308. This PtdIns(3,4)P_2_-mediated endosomal AKT activation provides a means to integrate PI3K activated by diverse stimuli to mTORC2 assembly. These early endosomal events induced by endocytosis, together with the previously identified AKT activation by PtdIns(3,4,5)P_3_, contribute to the strengthening of the transduction of AKT signaling through phosphoinositide.

## 1. Introduction

AKT/PKB is a central signaling effector that phosphorylates specific serine or threonine residues of core proteins that control cell physiological processes, such as cell survival, metabolism, growth, proliferation, and migration [1,2]. Since dysregulated AKT signaling is a major cause of human metabolic diseases, such as type II diabetes, obesity, and cancers, there are significant ongoing efforts to understand the upstream regulators of AKT. AKT activity is upregulated by phosphatidylinositol 3-kinase (PI3K) signaling during the activation of receptor tyrosine kinases or G-protein-coupled receptors [3], and full AKT activation in human cells requires the phosphorylation of Thr308 in the activation loop and Ser473 in the hydrophobic motif [4]. AKT is phosphorylated at Thr308 by phosphoinositide-dependent protein kinase 1 (PDK1) at the plasma membrane and at Ser473 by mechanistic target of rapamycin complex 2 (mTORC2) [5,6]. PDK1-mediated AKT Thr308 phosphorylation is dependent on the accumulation of PtdIns(3,4,5)P_3_ by PI3K at the plasma membrane, indicating that PtdIns(3,4,5)P_3_ molecules provide a binding platform for PDK1 and AKT through the pleckstrin homology (PH) domain of each protein.

Several studies showed AKT activation through phosphorylation at Ser473 via PtdIns(3,4,5)P_3_-dependent mTORC2 activation [7], via lysosome positioning of mTORC2 and AKT in response to serum [8], and via microtubule-associated protein 4-PI3K assembly and PtdIns(3,4,5)P_3_ production at endosomal compartments [9]. Nevertheless, the mechanism underlying mTORC2-dependent AKT activation through phosphorylation at Ser473 has not been elucidated based on cell types or their extracellular stimulations.

Mammalian target of rapamycin (mTOR) is a mammalian serine/threonine kinase and central regulator of cell growth and metabolism in response to nutrients and growth factors [10,11,12]. TOR is found in two structurally and functionally distinct protein complexes, TOR complex 1 (TORC1) and TOR complex 2 (TORC2) [13]. In mammalian cells, mTORC1 contains the following core components: mTOR; a regulatory protein associated with mTOR (Raptor); mammalian lethal with Sec13 protein 8 (mLST8), also known as GßL; and the two inhibitory subunits proline-rich AKT substrate of 40 kDa (PRAS40) and DEP domain-containing mTOR-interacting protein (DEPTOR) [14,15]. mTORC1 controls macromolecule biosynthesis, autophagy, cell cycle progression, growth, and metabolism [16,17]. On the other hand, mTORC2 comprises mTOR, rapamycin-insensitive companion of mTOR (Rictor), mammalian Sty1/Spc1-interacting protein (mSIN), mLST8, and Protor1/2. mTORC2 is activated by growth factors and controls cell survival, metabolism, and the cytoskeleton via several members of the AGC kinase subfamily, including AKT, serum- and glucocorticoid-induced protein kinase 1 (SGK1), and protein kinase C-α (PKC-α) [18,19,20]. According to previous studies, mTORC1 activation depends on proteins, including the GTPases RAG and RHEB, for lysosomal recruitment and activation [16]. In contrast, the upstream regulatory mechanisms of mTORC2 activation by growth factors have remained largely unknown [10,13]. In mammalian cells activated by growth factors, active mTORC2-AKT is associated with the ribosome [21], with the lysosome [8], or mitochondria-associated ER membrane (MAM), which is a subdomain of the ER tethered to mitochondria [22]. The mechanism by which growth factor stimulation controls the localization of active mTORC2 has yet to be elucidated. In yeast, TORC2 localization in the specific domain of the plasma membrane is essential for cell survival [23], but little is known about the upstream signals.

Furthermore, during activation with growth factors, the conversion of PtdIns(4,5)P_2_ into PtdIns(3,4,5)P_3_ by PI3K activated at the microdomain of the plasma membrane is critical for AKT signal transduction. A transient increase in PtdIns(3,4,5)P_3_ molecules provides the docking site at the plasma membrane for PDK1 [24] and AKT [25] through each protein’s PH domain. mTORC2 activity regarding AKT phosphorylation was found to be dependent on PI3K activation by growth factors [5,26]. A recent study using intracellular compartment-specific reporters suggested that mTORC2 activity does not depend on plasma membrane PI3K and is associated with endomembrane compartments, such as endosomes [27]. Another study proposed that the binding of PtdIns(3,4,5)P_3_ to the PH domain of SIN1, a component of mTORC2, activates mTORC2 kinase by inducing a conformational change from an inactive complex to an active one [7]. However, several studies demonstrate AKT phosphorylation and activation by mTORC2 is largely dependent on PtdIns(3,4)P_2_ molecules than on PtdIns(3,4,5)P_3_ molecules [28,29]. Therefore, whether the molecules connecting mTORC2 and AKT are phosphorylated phosphatidylinositols must be clarified in addition to the localization of mTORC2-induced AKT phosphorylation in the cell.

Herein, we show that AKT activation via Ser473 phosphorylation by mTORC2 requires endocytosis in U87MG glioma cells during platelet-derived growth factor (PDGF) stimulation. We found that mTORC2 is recruited into the early endosome using confocal imaging and biochemical approaches with endosomal fractionation in PDGF-activated cells. We also demonstrate that mSIN1—an essential contributor for mTORC2 activity and integrity—is localized to the early endosome in a PtdIns(3,4)P_2_-dependent manner and that endosomal mTORC2 is responsible for AKT activation through Ser473 phosphorylation. This study highlights the signaling role of PtdIns(3,4)P_2_ molecules in early endosomes promoted by endocytosis for mTORC2-mediated AKT activation.

## 2. Materials and Methods

### 2.1. Cell Culture, Transfection, and Reagents

U87MG cells (ATCC, Manassas, VA, USA) were cultured in DMEM (Welgene, Gyeongsan, South Korea, and PAN Biotech, Aidenbach, Germany) supplemented with 10% fetal bovine serum (Gibco-BRL, Grand Island, NY, USA and PAN Biotech) and 1% penicillin–streptomycin (HyClone, Logan, UT, USA, and PAN Biotech). Cells were grown at 37 °C in a humidified atmosphere of 5% CO_2_. Mycoplasma testing was conducted periodically. Cells were subjected to transient transfection with expression plasmids using the Effectene Reagent (QIAGEN, Hilden, Germany) or the Neon transfection method (Thermo Fisher Scientific, Waltham, MA, USA), according to the manufacturers’ manual. The following reagents were used at doses indicated in the figure legends: Rapamycin were obtained from Invivogen, Hong Kong; recombinant rat PDGF–BB (520-BB-050) was purchased from R&D Systems, Minneapolis, MN, USA; chlorpromazine was purchased from Sigma-Aldrich, St. Louis, MO, USA; Pitstop 2 was purchased from Abcam, Cambridge, United Kingdom; and CellMask Plasma Membrane Stain was purchased from Life Technologies, Carlsbad, CA, USA.

### 2.2. Plasmids and siRNAs

Human pMSCV-SIN1.1-Myc (plasmid12576) was obtained from David Sabatini through the Addgene repository. Full-length mSIN1 cDNA was amplified using PCR. The PCR primer sequences for the reactions were 5′-AATGGCCGGCCAATGGCCTTCTTGGACAATCCAACTATC-3′ (forward) and 5′-ATTGGCGCGCCTCACTGCTGCCCGGATTTCTTCTC-3′ (reverse). mSIN1-RBD-PH (RBD residues 276-354, PH residues 377-488) was amplified using PCR from pMSCV-SIN1.1-Myc. The PCR primer sequences for the reactions were as follows: 5′-TTGGCCGGCCGATGAAAGAGTCACTCTTTGTTCGAA-3′ (forward) and 5′-ATGGCGCGCCTCACGATTCCAGGATGTAGTTAAC-3′ (reverse). The amplified coding region was cloned into the Fse1 (GGCCGGCC) and Asc1 (GGCGCGCC) sites of the pCS_2_-Myc, pCS_2_-GFP, and pCS_2_-RFP vectors (cytomegalovirus promoter; provided by David Turner, University of Michigan Ann Arbor, MI, USA) for the expression of Myc, green fluorescent protein (GFP), and red fluorescent protein (RFP) epitope-tagged proteins, respectively. For the depletion of each phosphoinositide in early endosomes, the iRFP-FRB-Rab5 plasmid (51612; Addgene, Watertown, MA, USA) and mCherry-FKBP-MTM1 plasmid (51614; Addgene) were both provided by Tamas Balla (National Institutes of Health, Bethesda, MD, USA). The CFP-FKBP-Inp54p and the CFP-Rab5-Q79L (constitutively active Rab5) plasmids were obtained from Won Do Heo (KAIST, Daejeon, Korea). The mCherry-FKBP-INPP4B plasmid was generated using PCR amplification from the pEAK-Flag/INPP4B plasmid containing full-length INPP4B cDNA (24324; Addgene, USA) and cloning into the mCherry-FKBP plasmid. The Synaptojanin2 (SJ2)-Sac1-Rab5 plasmid was produced by replacing the cyan fluorescent protein (CFP) of CFP-Rab5 (provided by Won Do Heo) with PCR-amplified SJ2-Sac1 (the Sac1 domain of synaptojanin2) from the plasmid containing full-length SJ2 cDNA (kindly provided by Pietro De Camilli, Yale University, New Haven, CT, USA). The GFP-EEA1 plasmid (42307; Addgene) was provided by Silvia Corvera (University of Massachusetts Medical School, Worcester, MA, USA). Double-stranded siRNA oligonucleotides for mSIN1 were synthesized at Genolution (Seoul, South Korea) and were validated previously [30]. The target sequences were as follows: 5′-GCAGUCGAUAUUAUCUGUAUU-3′ and 5′-UACAGAUAAUAUCGACUGCUU-3′. A control siRNA for GFP (5′-GTTCAGCGTGTCCGGCGAGTT-3′) was obtained from Samchully Pharm (Seoul, South Korea). Synthetic siRNAs were introduced into cells by transfection using the NEON system (Thermo Fisher Scientific, USA).

### 2.3. Immunoblot Analysis and Antibodies

To obtain a total protein extract, cells were washed with cold phosphate-buffered saline (PBS), harvested, and sonicated with lysis buffer containing 25 mM HEPES-NaOH (pH 7.0), 2 mM EDTA, 25 mM β-glycerophosphate, 1% Triton X-100, 10% glycerol, a mixture of protease inhibitors (1 mM dithiothreitol [DTT], 5 mM NaF, 10 μg/mL aprotinin, and 10 μg/mL leupeptin), and a phosphatase inhibitor cocktail (Sigma-Aldrich). The lysates were centrifuged for 20 min at 12,000× *g* at 4 °C. The resulting supernatants were used for immunoblot analyses. Proteins were separated by sodium dodecyl sulfate–polyacrylamide gel electrophoresis (SDS–PAGE) and transferred to polyvinylidene fluoride (EMD Millipore, Burlington, MA, USA) or nitrocellulose (Thermo Fisher Scientific, USA) membranes. Specific signals were amplified by horseradish peroxidase-conjugated secondary antibodies (anti-rabbit IgG, 7074P2, Cell Signaling Technology, Danvers, MA, USA; goat-anti-mouse-IgG, SA001-500, genDEPOT, Barker, TX, USA) and visualized with WESTSAVE Up reagent ECL solution (Young In Frontier, Seoul, South Korea). Western blot images were quantified using Multi Gauge 3.0 software (Fujifilm, Japan). The following antibodies were used for immunoblotting: a rabbit polyclonal antibody specific for human peroxiredoxin II (PrxII) [31]; mSIN1 (05-1044, EMD Millipore; 1:1000), Rictor (ab56578, Abcam; 1:1000), GST (13-6700, Invitrogen; 1:500), EEA1 (610456, BD Transduction Laboratories, San Jose, CA, USA; 1:2000), HSP90 (610418, BD Transduction Laboratories; 1:1000), Rab5 (3547, Cell Signaling Technology; 1:1000), Rab7 (9367, Cell Signaling Technology; 1:1000), cathepsin D (ab6313, Abcam; 1:1000), mTOR (2983, Cell Signaling Technology; 1:1000), Raptor (2280, Cell Signaling Technology; 1:1000), GSK3β (9315, Cell Signaling Technology; 1:1000), phospho-AKT (Ser473) (9271, Cell Signaling Technology; 1:1000), phospho-AKT (Thr308) (2965, Cell Signaling Technology; 1:1000), AKT (9272S Cell Signaling Technology; 1:1000), phospho-GSK3β (Ser9) (9336, Cell Signaling Technology; 1:1000), phospho-TSC2 (Thr1462) (3617, Cell Signaling Technology; 1:1000), phospho-Fox01/3a/4 (Thr24/Thr32/Thr28) (2599, Cell Signaling Technology; 1:1000), phospho-p70 S6 Kinase (Thr389) (9205, Cell Signaling Technology; 1:1000), p70 S6 Kinase (9202, Cell Signaling Technology; 1:1000), Erk1/2 (9102, Cell Signaling Technology; 1:1000), and phospho-Erk1/2 (T202/Y204) (9101, Cell Signaling Technology; 1:1000). All Western blot experiments were independently repeated at least thrice. Full western blots can see Appendix A.

### 2.4. Analysis of PDGFR Endocytosis

PDGFR endocytosis was assessed as described previously [32], with some modifications. Serum-starved U87MG cells were incubated with bispecific scFv-Cκ-scFv fusion proteins (10 μg/mL) for 5 min at 37 °C for internalization. Cells were washed with cold PBS three times, then incubated with acidic buffer (0.2 M acetic acid, pH 2.7, 0.5 M NaCl) for 3 min at room temperature to remove antibodies bound to the cell surface. Cells were then washed with cold PBS twice, fixed in 4% paraformaldehyde (Electron Microscopy Sciences, Hatfield, PA, USA) in PBS for 10 min, and subjected to immunofluorescence staining analysis. Briefly, cells were incubated with PBS containing 5% horse serum (Gibco-BRL, USA) and 0.1% Triton X-100 for 30 min to block nonspecific antibody binding, and then with 2 μg/mL of a fluorescein isothiocyanate (FITC)-conjugated anti-human Cκ antibody (TB28-2, BD Biosciences, Franklin Lakes, NJ, USA) for 2 h at room temperature. Cells were also stained with 0.2 μg/mL 4′,6-diamidino-2-phenylindole (DAPI, Roche, Basel, Switzerland) to detect DNA. Confocal images were acquired with an LSM 880 microscope (Carl Zeiss, Oberkochen, Germany) at Ewha Fluorescence Core Imaging Center (Seoul, South Korea), and images were processed with Zen software (Carl Zeiss, Thornwood, New York, NY, USA).

### 2.5. In Vitro GST Pull-Down Assay for Purification of Rab5-GTP-Associated Endosomes

To purify Rab5-GTP-associated early endosomes, the Rab5-binding domain of the EEA1 effector of Rab5-GTP (EEA1-Rab5BD; residues 36–218, comprising C2H2 Zn^2+^ Finger (36–69) and coiled-coil domain (74–218), described and referenced in the Results) was amplified with PCR from the GFP-EEA1 plasmid and cloned into the pGEX-4T1 vector for GST protein fusion. The GST-EEA1-Rab5BD fusion protein was expressed in Escherichia coli strain BL21, purified according to the manufacturer’s instructions (GE Healthcare, USA), and stored at −70 °C. To isolate Rab5-positive endosomes, in vitro GST pull-down experiments were performed. In brief, after PDGF activation for 5 min, U87MG cells (3 × 10^6^) were collected and lysed in 800 μL lysis buffer (250 mM sucrose, 3 mM imidazole, pH 7.4, 0.5 mM EDTA, 1 mM DTT, 5 mM NaF, 10 μg/mL aprotinin, 10 μg/mL leupeptin, and a phosphatase inhibitor cocktail) by 10 passes through a 22 G1 1/4” needle fit on a 1-mL syringe. Broken cells were centrifuged for 20 min at 2000× *g* at 4 °C. Postnuclear supernatant (PNS) was collected for further analysis. GST-EEA1-RBD protein (4 μg) was added to PNS (800 μL) and incubated overnight at 4 °C to bind to Rab5-GTP-associated endosomes. The mixture was incubated for 2 h at 4 °C with 3 μL (binding capacity approximately 5 μg GST/μL) Glutathione-Sepharose-4B beads (GE Healthcare). After incubation, the beads were precipitated and washed five times with lysis buffer. Bound proteins were mixed with 2× SDS–PAGE loading buffer, resolved by SDS–PAGE, and immunoblotted to identify proteins.

### 2.6. Rapamycin (RAPA)-Mediated Depletion of Phosphoinositide in Endosomes

For depletion of each phosphoinositide from Rab5-GTP endosomes, U87MG cells were transiently transfected with iRFP-FRB-Rab5 (an early endosome anchoring protein), mCherry-FKBP-MTM1 (a RAPA-responsive targeting 3-phosphatase), mCherry-FKBP- INPP4B (a RAPA responsive targeting 4-phosphatase), or CFP-FKBP-Inp54p (a RAPA responsive targeting 5-phosphatase) for 24 h. Cells were serum starved for 5 h and then incubated with PDGF (50 ng/mL) in the presence of 40 nM RAPA for 5 min. To monitor the recruitment of each lipid phosphatase to the endosomes in the presence of RAPA, live-cell confocal microscopy was performed according to the following method. To monitor the amount of pAKT and PtdIns(3,4)P_2_, immunofluorescence analysis was carried out according to the following method with fixed cells.

### 2.7. Immunofluorescence and Live-Cell Confocal Imaging

Immunofluorescence analysis and live-cell confocal microscopy were performed as described previously, with some modifications [33]. Cells were cultured in 12-well dishes containing coverslips (diameter, 18 mm) coated with poly-L-lysine (Sigma-Aldrich, St. Louis, MO, USA) for both live-cell imaging and immunofluorescence staining. The cells were fixed for 10 min in 4% paraformaldehyde (Electron Microscopy Sciences) in PBS and incubated with a blocking solution containing 5% normal horse serum (Gibco-BRL, USA) and 0.1% Triton X-100 (Duchefa Biochemie, Haarlem, the Netherlands) in PBS for 30 min at room temperature. The cells were labeled with the primary antibody in a blocking solution for 30 min at room temperature or overnight on ice. After three washes with PBS for 5 min, the cells were incubated for 1 h at room temperature with a 1:1000 dilution of the fluorescence conjugated secondary antibody (Invitrogen) in blocking solution. After three washes with PBS, DAPI (0.2 μg/mL) was used for 2 min to detect DNA. Cells were mounted onto glass slides with Fluoromount-G (Southern Biotech, Birmingham, AL, USA). For coimmunostaining of other proteins, the cells were washed three times with PBS for 5 min and the blocking procedure was repeated. Phosphoinositide staining was performed as described previously [34]. Cells were fixed for 20 min in 2% paraformaldehyde and permeabilized with PBS containing 0.5% saponin (Sigma-Aldrich) and 1% bovine serum albumin (BSA, Bovogen, East Keilor, Australia) for 30 min at room temperature. Cells were incubated with lipid-specific antibodies for 2 h in PBS containing 1% BSA. After three 5-min washes with PBS, the cells were incubated with a fluorescent secondary antibody in PBS containing 1% BSA and washed three times for 5 min each with PBS. Confocal images were acquired using an LSM 880 (Carl Zeiss) or A1R (Nikon, Tokyo, Japan) confocal microscope. Live-cell images were obtained using the A1R microscope equipped with an incubation chamber (Live Cell Instrument, Namyangju, South Korea) at 37 °C in an atmosphere of 5% CO_2_. Images were processed using NIS-Elements (Nikon) or Zen (Carl Zeiss) software. The following antibodies were used for immunofluorescence staining: mSIN1 (05-1044, EMD Millipore; 1:200), Rictor (ab56578, Abcam; 1:330), Rab5 (3547, Cell Signaling Technology; 1:200), phospho-AKT (Ser473) (9271, Cell Signaling Technology; 1:25), phospho-AKT (Thr308) (13038, Cell Signaling Technology; 1:100), PtdIns(3,4)P_2_ (Z-P034b, Echelon Biosciences, Salt Lake City, UT, USA; 1:200), and PtdIns(3,4,5)P_3_ (Z-P345, Echelon Biosciences; 1:200).

### 2.8. Nikon-Structured Illumination Microscopy (N-SIM)

U87MG cells expressing GFP-mSIN were fixed with 4% paraformaldehyde and immunostained with antibodies to PtdIns(3,4)P_2_, followed by secondary antibodies conjugated with Alexa Fluor 546. SIM images were obtained using an N-SIM microscope (Nikon) equipped with a 63×/1.4 NA objective (1024 × 1024 pixels). To obtain line intensity histogram images, the acquired U87MG cell images were analyzed with NIS-Elements AR 4.2 software (Nikon).

### 2.9. AKT Kinase Assay

This assay was performed using an AKT kinase activity kit (ADI-EKS-400A; Enzo Life Sciences, Farmingdale, NY, USA) according to the manufacturer’s instructions with minor modifications. Serum-starved and PDGF-stimulated U87MG cells (2.5 × 10^6^ per 100-mm dish) were incubated in the absence or presence of chlorpromazine (5 μg/mL) or Pitstop2 (5 μM) for 3 min. The cells were then washed with cold PBS, harvested, and sonicated with 1 mL of lysis buffer containing 25 mM HEPES-NaOH, pH 7.0, 2 mM EDTA, 25 mM β-glycerophosphate, 1% Triton X-100, 10% glycerol, a mixture of protease inhibitors (1 mM DTT, 5 mM NaF, 10 μg/mL aprotinin, 10 μg/mL leupeptin, 1 mM Na_3_VO_4_, 1 mM phenylmethylsulfonyl fluoride), and a phosphatase inhibitor cocktail (Sigma-Aldrich). The supernatants collected by centrifugation at 12,000× *g* for 20 min at 4 °C were used for column chromatography using desalting columns with a molecular weight cutoff of 40,000 Daltons (87,769, Thermo Fisher Scientific) according to the manufacturer’s instructions. The resulting protein mixtures (molecular weight >40,000) were concentrated with an Amicon Ultra-2 centrifugal filter unit (UFC201024, EMD Millipore, USA) and used for the AKT kinase activity assay. The protein mixture (450 μg) was added to 96 wells of a microtiter plate containing the protein kinase B substrate. To initiate the reaction, ATP (final concentration 123 μM) was added to each well and the mixture was incubated for 90 min at 30 ℃. After three washes with PBS, the phospho-specific substrate antibody solution (40 μL) was added to each well and the mixture was incubated for 60 min at room temperature. After three washes with PBS containing 0.1% Triton X-100, each well containing the bound phospho-specific substrate antibody was incubated for 30 min at room temperature with a horseradish peroxidase-conjugated anti-rabbit IgG solution (40 μL, 1:1000). After three washes with PBS, each well was incubated for 45 min with 3,3′,5,5′-tetramethylbenzidine substrate solution (60 μL) to visualize the AKT activity signal. The reaction was stopped by adding HCl to a final concentration of 250 mM, and the signal was then measured using a SpectraMax M2 fluorescence microplate reader (Molecular Devices, San Jose, CA, USA) at an absorbance wavelength of 450 nm.

### 2.10. CellMask Plasma Membrane Staining

Serum-starved U87MG cells were treated with PDGF–BB (50 ng/mL) for 5 min and incubated with a 1:1000 dilution of CellMask deep red plasma membrane stain (C10046, Life Technologies) for 5 min at 37 °C. Cells were washed with PBS three times to remove the staining solution and fixed in 4% paraformaldehyde (Electron Microscopy Sciences) in PBS for 10 min. The cells were then subjected to immunofluorescence staining analysis.

### 2.11. Statistics and Reproducibility

All experiments were repeated independently at least three times. Quantitative data are provided as the means ± SEM of triplicate determinations from representative experiments. Quantitative data were analyzed with Student’s two-tailed *t*-test in Sigma Plot 10.0 and Excel in which a *p*-Value < 0.05 was considered statistically significant.

## 3. Results

### 3.1. AKT Activation via Phosphorylation at Ser473 Requires Endocytosis

Endocytosis during receptor activation is classically involved in cell desensitization during chronic receptor activation and also clearly promotes the signal propagating function in acute response to extracellular stimuli [35,36]. We investigated whether endocytosis is required for PDGF-induced AKT activation in U87MG glioma cells, which display hyperactivated AKT signaling caused by phosphatase and tensin homolog (PTEN) deficiency [37,38], using endocytosis inhibitors. The administration of the receptor recycling inhibitor chlorpromazine, which prevents coated pit assembly at the plasma membrane [39], or an inhibitor of clathrin-mediated endocytosis, Pitstop 2 [40], specifically decreased AKT phosphorylation at Ser473 but not at Thr308 (Figure 1). The AKT kinase assay with lysates from PDGF-stimulated cells in the presence of chlorpromazine or Pitstop 2 showed that these endocytosis inhibitors elicited a reduction in AKT activity by ~65% or ~30%, respectively (Figure 1E). These results indicate that endocytosis is critical for full AKT activation through Ser473 phosphorylation. To monitor the effect of chlorpromazine and Pitstop 2 on early endosome formation during PDGF-mediated endocytosis, we used bispecific monoclonal antibodies to PDGF receptor (PDGFR)β, as described in our previous work [32]. The inhibitory effect of chlorpromazine and Pitstop 2 on endosome formation was verified using immunofluorescence staining and monitoring endocytosed fluorescently labeled PDGFRs taking advantage of bispecific anti-mPDGFRβ × cotinine scFv-Cκ-scFv fusion protein (Appendix A). Confocal imaging showed the newly formed endosomes, which come from receptor-mediated endocytosis after removal of fluorescently labeled antibodies against PDGFRs, unendocytosed on the plasma membrane surface by washing with an acetic acid buffer (pH = 2.7) and a reduction in the early endosome numbers by chlorpromazine and Pitstop 2 in a concentration-dependent manner. Growth factor-induced endocytosis plays a critical role in sustained epidermal growth factor (EGF) receptor signaling on AKT [41] and in the establishment and control of specific downstream signal transduction pathways [42]. Based on our results, PDGFRβ-mediated endocytosis is critical to AKT activation via phosphorylation at Ser473 which led us to investigate the mTORC2 assembly at the early endosome during PDGF stimulation.

### 3.2. mTORC2 Is Recruited into the Early Endosome upon Growth Factor Stimulation

Considering that mTORC2 is responsible for AKT phosphorylation at Ser473 [5,6,24], we next investigated whether key components of mTORC2 are localized in early endosomes. First, we analyzed the localization of mSIN, which is required for mTORC2 assembly and integrity [43], using immunofluorescence imaging. Many spots corresponding to mSIN staining colocalized with early endosomes identified by endocytosed PDGFRβ in U87MG glioma cells (Figure 2A). The number of Rab5-positive puncta indicating early endosomes increased in cells stimulated with PDGF, compared to starved cells (Figure 2B), similar to the findings of a previous report on EGF-stimulated cells [44]. Colocalization of mSIN and Rab5 demonstrates that mSIN recruitment to early endosomes depends on growth factor activation (Figure 2B), indicating that PDGFRβ activation results in the promotion of mTORC2 assembly in early endosomes. mSIN contains a Ras-binding domain (RBD, residues 276–354 in SIN1.1) and a PH domain (residues 377–488 in SIN1.1), which provides a putative binding site for phosphatidylinositol polyphosphates [45]. Therefore, we monitored the localization of GFP-mSIN-RBD-PH (residues 276–488 in SIN1.1) in U87MG glioma cells under PDGF–BB activation using confocal live-cell imaging (Figure 2C). We found that GFP-mSIN-RBD-PH-positive puncta increased by ~50% near the plasma membrane in the presence of PDGF (Figure 2D and Appendix A). We verified that GFP-mSIN-RBD-PH-positive puncta correspond to early endosomes containing constitutively active GTPase (Rab5-Q79L) using a colocalization experiment (Figure 2E). Imaging showed that Rictor, an obligate component of mTORC2, also partially localized to early endosomes (Rab5-positive endosomes [46]), indicating that active mTORC2 originates in early endosomes during growth factor activation (Figure 2F).

Moreover, we used a biochemical approach for the purification of early endosomes to assess whether components of mTORC2 exist in these cellular organelles in the absence or presence of PDGF–BB. To isolate early endosomes containing Rab5-GTP, we purified recombinant GST-EEA1-Rab5-GTP-binding domain (GST-EEA1-Rab5BD, residues 36–218 in EEA1, comprising C2H2 Zn^2+^ Finger (residues 36–69) and coiled-coil domain (residues 74–218)) from bacteria. The human EEA1 C2H2 Zn^2+^ finger domain was previously identified as an active Rab5A-binding region [47]. We added GST-EEA1-Rab5BD proteins to the lysates of U87MG cells before and after PDGF activation, for 5 min each time, and isolated active Rab5 endosomes with glutathione (GSH) beads (Figure 3A). The immunoblot analysis of purified early endosomal fractions revealed that mSIN, Rictor, and mTORC—components of mTORC2—were associated with Rab5-GTP-bound endosomes but not Raptor, a component of mTORC1 (Figure 3B–F). In addition, phosphorylated AKT at Ser473 via mTORC2 was observed in early endosomes. Late endosome-specific Rab7 and lysosome-resident cathepsin D were not detected in early endosomes purified with the Rab5-GTP-binding domain. This result shows that subcellular AKT activation by early endosome-associated mTORC2 is distinct from that by mTORC2 at lysosomes and late endosomes during growth factor stimulation [8] and by lysosome-dependent mTORC1 in response to various stimuli [17,48]. Furthermore, confocal imaging showed that most puncta corresponding to mSIN or Rictor did not colocalize with the plasma membrane (Appendix A), indicating mTORC2 assembly in early endosomes.

### 3.3. Localization of mTORC2 to Early Endosomes Is Mediated by PtdIns(3,4)P_2_

Next, we investigated whether phosphatidylinositol polyphosphate mediates mTORC2 translocation into the early endosome. Previous studies proposed the distinct role of PtdIns(3,4)P_2_-dependent AKT phosphorylation at Ser473, in addition to PtdIns(3,4,5)P_3_-dependent AKT phosphorylation at Thr308 in AKT activation [28,29,49]. PI3K recruited to the plasma membrane by activated receptor tyrosine kinases sequentially generates PtdIns(3,4,5)P_3_ and PtdIns(3,4)P_2_, and the accumulation of these molecules at the plasma membrane induces AKT phosphorylation at Thr308 and Ser473, respectively [28,29]. In turn, the dual phosphorylation of AKT at these residues contributes to its full activity [28,29,50]. Considering that AKT Ser473 is phosphorylated by mTORC2 [5] and blocking mTORC2 activity by genetic ablation of mSIN1 or Rictor—both of which are mTORC2-specific components—affects AKT substrate specificity [43,51], the physiological consequence of AKT activity regulation by mTORC2 must be further investigated. Cytosolic AKT proteins are recruited to and activated near the microdomain of the plasma membrane via the interaction of the AKT-PH domain with PtdIns(3,4,5)P_3_ or PtdIns(3,4)P_2_ [25,52]. To examine how mTORC2 is recruited into early endosomes, we compared the localization of mSIN-RBD-PH with that of AKT-PH using live-cell confocal imaging (Figure 4A and Appendix A). The fluorescence intensity of AKT-PH, which can detect both PtdIns(3,4,5)P_3_ and PtdIns(3,4)P_2_, increased both at the plasma membrane and in the cytosol close to the inner leaflet of the plasma membrane Intriguingly, mSIN-RBD-PH signals colocalized with some AKT-PH signals only, exhibiting a spotty pattern inside the plasma membrane in a representative image of a live U87MG cell during PDGF–BB stimulation (Figure 4A). Given that PtdIns(3,4)P_2_ generation lags behind PtdIns(3,4,5)P_3_ during extracellular stimulation [53,54] and that PtdIns(3,4)P_2_ is produced by SH2-containing inositol phosphatase (SHIP) family proteins from PtdIns(3,4,5)P_3_ after PI3K activation [55], we next investigated whether mSIN colocalizes to the early endosomes containing PtdIns(3,4)P_2_ in PDGF-activated cells using an antibody against PtdIns(3,4)P_2_. The specificity of the antibody to PtdIns(3,4)P_2_ was verified by an increase in the amount of PtdIns(3,4)P_2_ around the plasma membrane in a PDGF-dependent manner and by the overlapping of the signal of the antibody with that of a known fluorescent PtdIns(3,4)P_2_ indicator, GFP-TAPP1-PH [56] (Appendix A). PtdIns(3,4)P_2_ colocalized with mSIN, a component of mTORC2, and Rab5, an early endosomal marker, as puncta inside the plasma membrane of PDGF-activated U87MG cells (Figure 4B–D). Structured illumination microscopy (SIM) revealed that mSIN colocalized with PtdIns(3,4)P_2_ at a distance of 30–60 nm, thereby indicating the regions of mSIN that lack PtdIns(3,4)P_2_ (Figure 4E). We rarely observed PtdIns(3,4,5)P_3_ in Rab5-positive endosomes and their colocalization with mSIN in PDGF–BB-stimulated cells (Appendix A).

### 3.4. Forced Depletion of Endosomal PtdIns(3,4)P_2_ by Targeted Phosphatases Causes Reduced AKT Phosphorylation at Ser473

The presence of mTORC2 in early endosomes motivated us to test the possibility that the endosomal pool of mTORC2 is required for AKT activation and depends on the accumulation of PtdIns(3,4)P_2_. To this end, we investigated if the specific removal of endosomal phosphoinositide by the rapamycin-inducible targeting of phosphatases into early endosomes affects AKT activation. We selected myotubularin 1 (MTM1) as a PtdIns 3-phosphatase [57], INPP4B as a PtdIns 4-phosphatase [58], and Inp54p as a PtdIns 5-phosphatase [59]. Furthermore, we used the rapamycin-dependent heterodimerization of two components (iRFP-FRB-Rab5 and mCherry-FKBP-MTM1 for 3-phosphatase recruitment; iRFP-FRB-Rab5 and mCherry-FKBP-INPP4B for 4-phosphatase recruitment; iRFP-FRB-Rab5 and CFP-FKBP-Inp54p for 5-phosphatase recruitment) of the system with some modifications, as previously described [60], to selectively deplete each phosphoinositide in endosomes (Figure 5A). The recruitment of each lipid phosphatase (MTM1, INPP4B, and Inp54p) into early endosomes in the presence of rapamycin was verified using confocal microscopy (Appendix A). The level of AKT Ser473 phosphorylation in early endosomes was reduced by ~50% in cells expressing endosome-targeting MTM1 (3-phosphatase) or INPP4B (4-phosphatase), but in cells expressing Inp54p (5-phosphatase), it remained unaltered in the presence of rapamycin during PDGF–BB stimulation (Figure 5B,C). Image quantification shows that depletion of PtdIns(3,4)P_2_, an endosomal phosphoinositide containing two phosphates at the 3rd and 4th position of the inositol ring, inhibited local AKT activation through Ser473 phosphorylation. This result indicates that AKT activation via mTORC2 depends on PtdIns(3,4)P_2_, but not PtdIns(3,4,5)P_3_, at the early endosome, which confirms findings from a previous report [27] that mTORC2 is targeted to the cellular membrane compartment in a PtdIns(3,4,5)P_3_-independent manner. Since the level of AKT Thr308 phosphorylation remained consistent in cells expressing each lipid phosphatase in the presence of rapamycin (Figure 5D), we speculate that AKT activation by Thr308 phosphorylation does not depend on endosomal phosphoinositide and results predominantly from the accumulation of PtdIns(3,4,5)P_3_ at the plasma membrane [24,61]. We verified that the rapamycin-induced localization of 3-phosphatase or 4-phosphatase to early endosomes results in reduced endosomal PtdIns(3,4)P_2_ but that of 5-phosphatase dose not (Figure 5E,F). The activity of FKBP-MTM1, FKBP-INPP4B, and FKBP–Inp54p has been well demonstrated in previous works [60,62,63,64,65]. We investigated if rapamycin treatment (40 nM, 5 min incubation) for lipid phosphatase targeting may indirectly affect AKT activation in our experiments, as it was shown that rapamycin inhibits mTORC1 activity acutely [66] and mTORC2 activity chronically [67]. AKT phosphorylation of both Ser473 and Thr308 remained unaltered by rapamycin treatment during the 5 min PDGF stimulation (Appendix A). Taken together with the presence of mTORC2 components in early endosomes (Figure 2, Figure 3 and Figure 4), the targeting of specific phosphoinositide phosphatases with the rapamycin-dependent heterodimerization strongly indicates that AKT activation by Ser473 phosphorylation requires an increase in the amount of PtdIns(3,4)P_2_ and mTORC2 in early endosomes of cells activated with growth factors.

### 3.5. Endosomal Phosphoinositide Accumulation Is Required for the Activation of a Subset of AKT Substrates

Next, we investigated the effect of endosomal phosphoinositide accumulation on the activation of various AKT substrates. We transfected U87MG cells with a vector encoding a modified form of the Sac1 domain (Sac1) of synaptojanin 2 (SJ2-Sac1) as a PtdIns 4-phosphatase [68] with a Rab5 GTPase (SJ2-Sac1-Rab5) or encoding CFP-Rab5 as a control. The immunoblot analysis of lysates from transfected U87MG cells showed that the level of pAKT(Ser473) in cells expressing endosome-targeting SJ2-Sac1-Rab5 decreased in an expression-dependent manner compared to cells expressing CFP-Rab5 (Figure 6A,B). In contrast, the level of pAKT(Thr308) remained unchanged in both cell lines during stimulation with PDGF. The results indicate that phosphoinositides containing phosphate at the fourth position of the inositol ring, such as PtdIns(4)P, PtdIns(3,4)P_2_, and PtdIns(3,4,5)P_3_, can be involved in endosomal AKT phosphorylation and activation. Moreover, high expression of SJ2-Sac1-Rab5 essentially inhibited PDGF-induced phosphorylation of a subset of AKT substrates, including glycogen synthase kinase 3 beta (GSK3β) at Ser9, forkhead box protein O1 (FoxO1) at Thr24, and forkhead box protein O3a (FoxO3a) at Thr32, by ~60% and tuberous sclerosis complex 2 (TSC2) by ~20%, whereas the level of pS6K(Thr389), an mTORC1 substrate, was unaltered in cells expressing SJ2-Sac1-Rab5 (Figure 6C–F). We also observed that ectopic expression of SJ2-Sac1-Rab5 in early endosomes caused a reduction in PtdIns(3,4)P_2_ and mSIN by ~60% and ~70%, respectively, compared to CFP-Rab5 (Appendix A), indicating that endosomal PtdIns(3,4)P_2_ is critical for mTORC2 association with early endosomes. A reduction in the levels of pAKT(Ser473), pGSK3β(Ser9), and total GSK3β was observed in mSIN knock-down cells compared to siCont-treated cells (Appendix A).

## 4. Discussion

In this study, we first demonstrated that in PDGF-activated cells, endocytosis is required for AKT phosphorylation at Ser473 but dispensable for AKT phosphorylation at Thr308. Given that the sequential phosphorylation of AKT at both Thr308 and Ser473 is critical for the fidelity of AKT signal transduction and the specificity of downstream substrate phosphorylation [28,29,43,50,51], this finding supports the fact that the signaling endosomes originate from endocytosis and provide platforms that enable intracellular signal propagation and maintenance [35,36,41]. Secondly, we showed that PtdIns(3,4)P_2_ molecules are key factors for the interaction of AKT with mTORC2 via mSIN in early endosomes. Therefore, local mTORC2 activity in early endosomes is responsible for AKT Ser473 phosphorylation and the activation of a subset of AKT substrates, such as GSK3β and FoxO1/3a. In glioma cells, we demonstrate that an increase in endosomal PtdIns(3,4)P_2_ in response to PDGF–BB is a prerequisite for AKT phosphorylation at Ser473 but not at Thr308 using a drug-inducible heterodimerization system and for targeting mTORC2 into endosomes via mSIN, which contains an RBD-PH domain, to specifically bind PtdIns(3,4)P_2_. Thirdly, we highlighted the consequence of PDGF-induced AKT activation in endosomes during endocytosis. Newly formed signaling endosome induced AKT activation results in the increased specificity by inducing the phosphorylation of GSK3β and FoxO1/3a without affecting mTORC1 signaling. Here, we also explore the fact that the control of AKT signaling in subcellular compartments, such as ribosomes, MAMs, and lysosomes, was reported to have advantages to orchestrate growth and metabolism [8,21,22].

A model illustrating the mechanism underlying PtdIns(3,4)P_2_ accumulation in early endosomes and its role in AKT signaling is shown in Figure 7.

According to our proposed model, an activated PDGF receptor induces PtdIns(3,4,5)P_3_ production from PtdIns(4,5)P_2_ by activating class I PI3K. In turn, increased PtdIns(3,4,5)P_3_ promotes AKT Thr308 phosphorylation by direct association of AKT with PDK1 through interaction with a PH domain. SHIP family 5-phosphatases are likely to produce PtdIns(3,4)P_2_ from PtdIns(3,4,5)P_3_ near the plasma membrane [69,70]. In PTEN-deficient U87MG glioma cells, PtdIns(3,4)P_2_ production predominantly depends on the high PtdIns(3,4,5)P_3_ accumulation, according to a study using a high-avidity biosensor of PtdIns(3,4)P_2_ [62], which is distinct from class II PI3K-driven PtdIns(3,4)P_2_ production from PtdIns(4)P during clathrin-mediated endocytosis [34]. Early endosomal PtdIns(3,4)P_2_ especially binds to the PH domains of AKT and mSIN1. Endosome-associated AKT displays an open conformation via allosteric binding to PtdIns(3,4)P_2_ [61], and neighboring mTORC2 on the platform enriched with PtdIns(3,4)P_2_ molecules further activates AKT through Ser473 phosphorylation in endosomes. In turn, AKT promotes the phosphorylation of GSK3β and FoxO1/3a to transduce specific AKT signals. Our model supports the findings of previous reports showing that the endosomal compartment of AKT and its selected effectors are important for the specificity of signal transduction [50,71].

In this study, we propose that endosomal PtdIns(3,4)P_2_ molecules play a key role in the mTORC2 mediated activation of AKT. AKT is most frequently hyperactivated in human cancers by mutations affecting upstream regulators, but AKT inhibitor treatment use has been restricted as the direct inhibition of AKT activity can lead to high cytotoxicity. PtdIns(3,4)P_2_ is predominantly produced by class I PI3 kinase and SHIP 5-phosphatase activated sequentially from PtdIns(4,5)P_2_ during growth factor stimulation [62]. Our study provides a rationale for a different approach to inhibit mTORC2 activity via reducing the endosomal PtdIns(3,4)P_2_ level for AKT-targeting therapeutic agents with minimal side effects by avoiding the inhibition of the downstream AKT signaling broadly.

## 5. Conclusions

In glioma cells, we demonstrate AKT activation in early endosomes during PDGF stimulation. mTORC2, which is associated with AKT activation, is recruited into endosomes by PtdIns(3,4)P_2_. We also show that endocytosis is key to AKT activity and endosomal AKT activation regulates AKT substrates such as GSK3β and FoxO1/3.

## Figures and Tables

**Figure 1 cancers-13-02405-f001:**
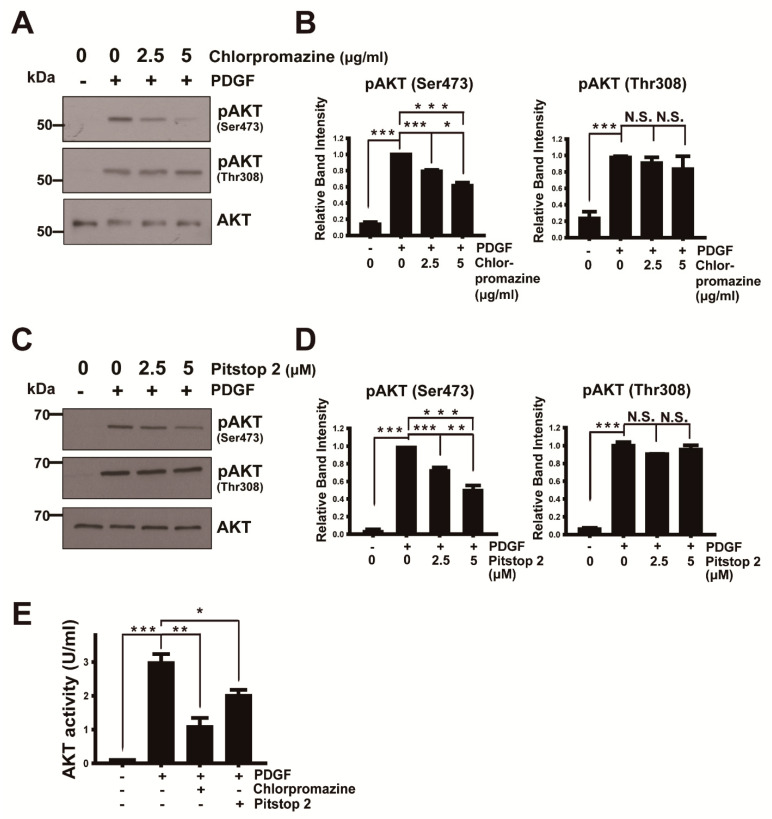
Endocytosis is critical for acute phosphorylation of AKT at Ser473 but not at Thr308: (**A**) serum-starved U87MG cells were pretreated with chlorpromazine (2.5 μg/mL, 5 μg/mL) for 3 min and incubated with platelet-derived growth factor-BB (PDGF–BB) (5 ng/mL) in the presence or absence of chlorpromazine for 3 min. Cell lysates were subjected to immunoblot analysis with antibodies specific to the indicated proteins; (**B**) the relative immunoblot signal intensity of pAKT(Ser473) and pAKT(Thr308) normalized over total AKT was determined as the mean ± SEM from three independent experiments, and *p*-values were calculated using Student’s two-tailed *t*-test. * *p* < 0.05, *** *p* < 0.001, N.S., not significant; (**C**) serum-starved U87MG cells were pretreated with Pitstop 2 (2.5 μM, 5 μM) for 5 min and incubated with PDGF–BB (5 ng/mL) in the presence or absence of Pitstop 2 for 3 min. Cell lysates were subjected to immunoblot analysis with antibodies specific to the indicated proteins; (**D**) the relative immunoblot signal intensity of pAKT(Ser473) and pAKT(Thr308) normalized over total AKT was determined as the mean ± SEM from three independent experiments, and *p*-values were calculated using Student’s two-tailed *t*-test. ** *p* < 0.01, *** *p* < 0.001, N.S., not significant; (**E**) serum-starved U87MG cells were pretreated with chlorpromazine (5 μg/mL) for 3 min or Pitstop 2 (5 μM) for 5 min and incubated with PDGF–BB (5 ng/mL) in the presence or absence of chlorpromazine (5 μg/mL) or Pitstop 2 (5 μM) for 3 min. Cell lysates were assayed for AKT kinase activity using the AKT Kinase Activity Kit (Enzo Life Sciences). Data are presented as the mean ± SEM from three experiments, and *p*-values were calculated using Student’s two-tailed *t*-test. * *p* < 0.05, ** *p* < 0.01, *** *p* < 0.001.

**Figure 2 cancers-13-02405-f002:**
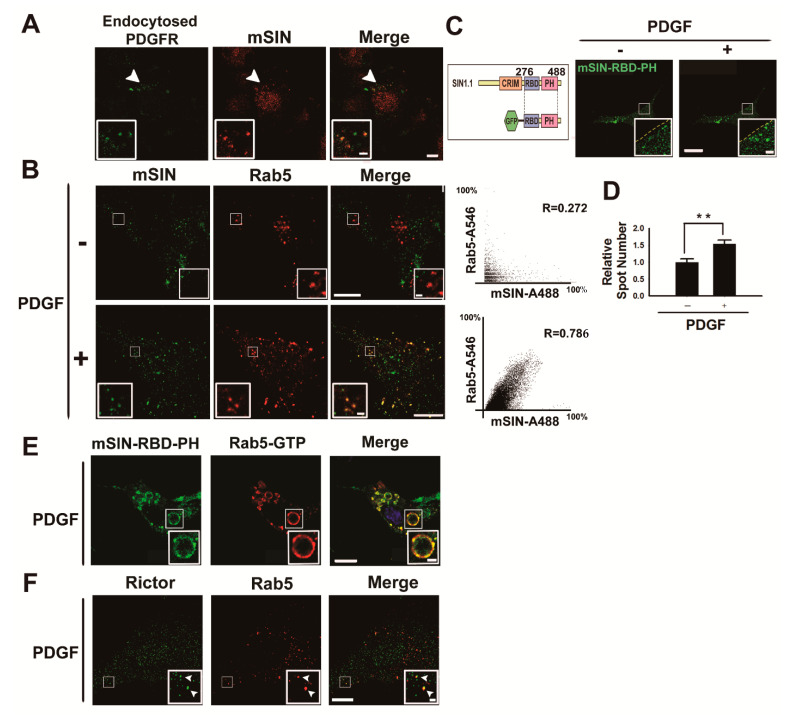
mTORC2 localizes to the early endosome during growth factor activation: (**A**) confocal microscopy of U87MG cells activated with bispecific anti-mPDGFRβ × cotinine scFv-Cκ-scFv fusion protein (10 μg/mL, green) for 5 min and stained with an antibody against mSIN (red). The experimental procedure is described in the Materials and Methods. mSIN colocalized with endocytosed bispecific scFv-Cκ-scFv fusion proteins. The areas indicated by arrowheads are shown at higher magnification in the insets. Scale bar, 10 μm; insets, 2 μm; (**B**) confocal microscopy of U87MG cells activated with PDGF–BB (50 ng/mL) for 5 min and stained with antibodies against Rab5 (red) and mSIN (green). Pearson correlation of mSIN-A488 and Rab5-A546 relative fluorescence intensity in the absence (PDGF–) and presence (PDGF+) of PDGF–BB is shown in the plot. Many endogenous mSIN puncta colocalized with endogenous Rab5 in cells activated with PDGF–BB. Scale bars, 20 μm; insets, 2 μm; (**C**) images of live U87MG cells before (left) and at 4.5 min (right) after PDGF–BB (50 ng/mL) treatment. Live images were obtained from U87MG cells expressing GFP-mSIN-RBD-PH (residues 276–488 in SIN1.1) using a laser-scanning confocal microscope (Nikon A1R) at 30 s intervals for 15 min. The areas indicated by boxes are shown at higher magnification in the insets. Scale bars, 20 μm; insets, 2 μm; (**D**) quantification of the relative spot number of (**C**). Data are presented as the mean ± SEM, and *p*-values were calculated using Student’s two-tailed *t*-test. ** *p* < 0.01 (*n* = 6 cells). The areas indicated by boxes are shown at higher magnification in the insets. Scale bar, 10 μm; insets, 1 μm; (**E**) confocal microscopy of U87MG cells expressing GFP-mSIN-RBD-PH (green) and CFP-Rab5-Q79L (red, constitutively active Rab5) during PDGF–BB (50 ng/mL) activation for 5 min stained with antibodies to Rab5. The fluorescence signal of mSIN-RBD-PH was localized close to that of CFP-Rab5-Q79L in giant early endosomes. The areas indicated by boxes are shown at higher magnification in the insets. Scale bars, 20 μm; insets, 1 μm; (**F**) confocal microscopy of U87MG cells activated with PDGF–BB (50 ng/mL) for 5 min and stained with antibodies against Rab5 (red) and Rictor (green). The areas indicated by boxes are shown at higher magnification in the insets. The arrowheads show the overlap of Rictor and Rab5. Scale bar, 20 μm; insets, 1 μm.

**Figure 3 cancers-13-02405-f003:**
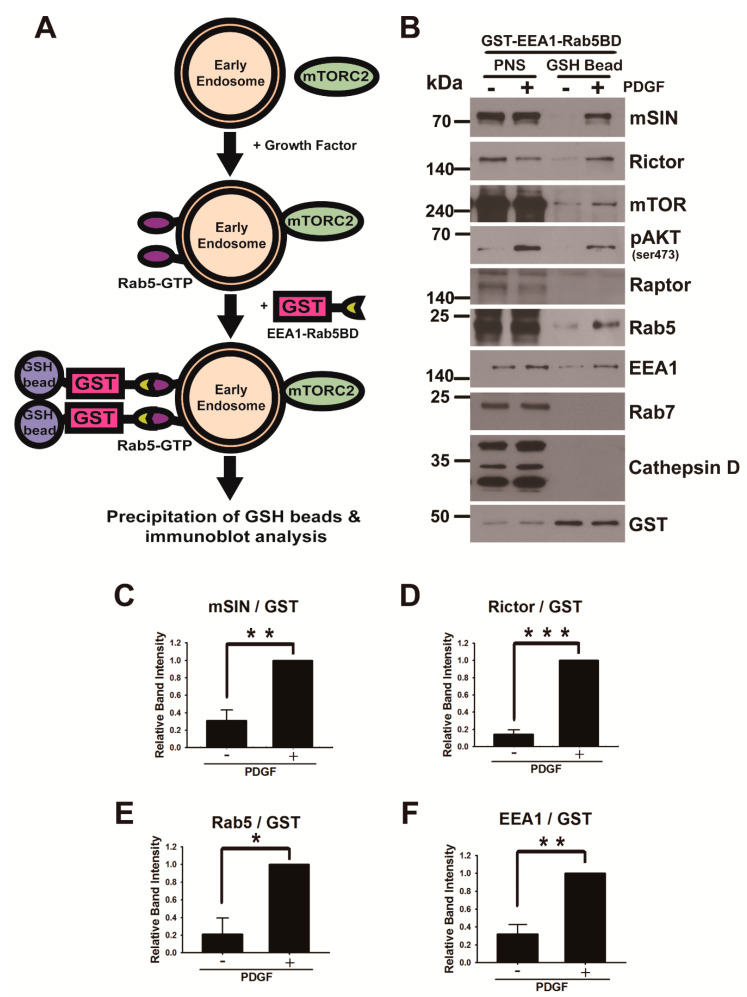
mTORC2 is associated with early endosomes: (**A**) proposed scheme to assess the association of mTORC2 with early endosomes. Endosomes containing Rab5-GTP (early endosomes) were added to recombinant GST-EEA1-Rab5BD proteins and purified by pull-down of GSH beads. Proteins eluted from beads were analyzed by immunoblots; (**B**) early endosomes from lysates of serum-starved U87MG cells (PDGF–) and cells activated with PDGF–BB (50 ng/mL) for 5 min (PDGF+) were prepared according to the Materials and Methods. To purify endosomes containing Rab5-GTP (early endosomes), we added GST-EEA1-Rab5BD recombinant proteins to the postnucleus supernatant (PNS), and GSH-bead-bound fractions were precipitated. PNS and GSH-bead precipitates were subjected to immunoblot analysis with antibodies specific to the indicated proteins; (**C**–**F**) The relative immunoblot signal intensity of GSH-bead-bound mSIN, Rictor, Rab5, and EEA1 (a marker for Rab5-positive endosomal fractions) normalized by that of GST were determined as the mean ± SEM from three independent experiments, and *p*-values were calculated using Student’s two-tailed *t*-test. * *p* < 0.05, ** *p* < 0.01, *** *p* < 0.001. Two components (mSIN and Rictor) of mTORC2 were associated with early endosomes.

**Figure 4 cancers-13-02405-f004:**
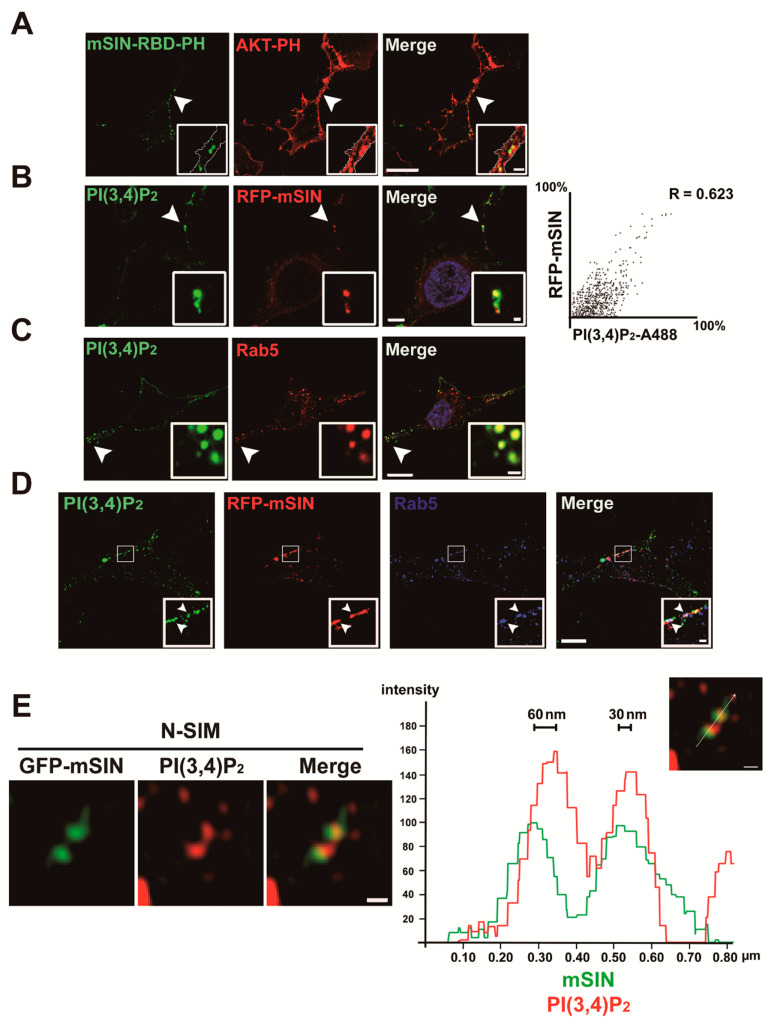
mTORC2 is associated with PtdIns(3,4)P_2_ molecules in early endosomes: (**A**) serum-starved U87MG cells expressing GFP-mSIN-RBD-PH and mCherry-AKT-PH (a reporter for PtdIns(3,4,5)P_3_ and PtdIns(3,4)P_2_) were activated with PDGF–BB (50 ng/mL) for 15 min for live-cell imaging. A representative image from a PDGF-stimulated live cell shows the partial overlap of both signals (green and red). The areas indicated by arrowheads are shown at higher magnification in the insets. Magnified images show that mSIN-RBD-PH (green) colocalizes with the intracellular portion of mCherry-AKT-PH (red) near the plasma membrane. The colocalized areas were presumed to be early endosomes inside the plasma membrane. The white line in the insets indicates the edge of a cell. Scale bars, 20 μm; insets, 2 μm; (**B**–**D**) confocal microscopy of U87MG cells activated with PDGF–BB (50 ng/mL) for 5 min and stained with antibodies specific to PtdIns(3,4)P_2_ (green) and Rab5 (red in (**C**) and orange in (**D**)). RFP-mSIN cDNA was transfected in (**B**,**D**) to avoid the overlap of primary antibodies from the same host species and imaged by a red 561 nm laser. Pearson correlation of PtdIns(3,4)P_2_-A488 and RFP-mSIN relative fluorescence intensity is shown in (**B**). PtdIns(3,4)P_2_ molecules colocalized with mSIN and Rab5. The areas indicated by arrowheads (**B**,**C**) and boxes (**D**) are shown at higher magnification in the insets (**B**–**D**). The arrowheads in the insets of (**D**) show the overlap of PtdIns(3,4)P_2_ (green), mSIN (red), and Rab5 (blue). Scale bars, 10 μm, 1 μm (insets in (**B,D**)), 0.5 μm (insets in (**C**)). (**E**) N-SIM of U87MG cells activated with PDGF–BB (50 ng/mL) for 5 min and stained with an antibody specific to PtdIns(3,4)P_2_ (red). GFP-mSIN cDNA was transfected to avoid the overlap of primary antibodies from the same host species and imaged by a green 488 nm laser. PtdIns(3,4)P_2_ molecules colocalized with mSIN. Scale bars, 0.2 μm. The fluorescence intensity histogram shows the localization of mSIN (green) and PtdIns(3,4)P_2_ (red). Data shown are from a single representative experiment out of three repeats.

**Figure 5 cancers-13-02405-f005:**
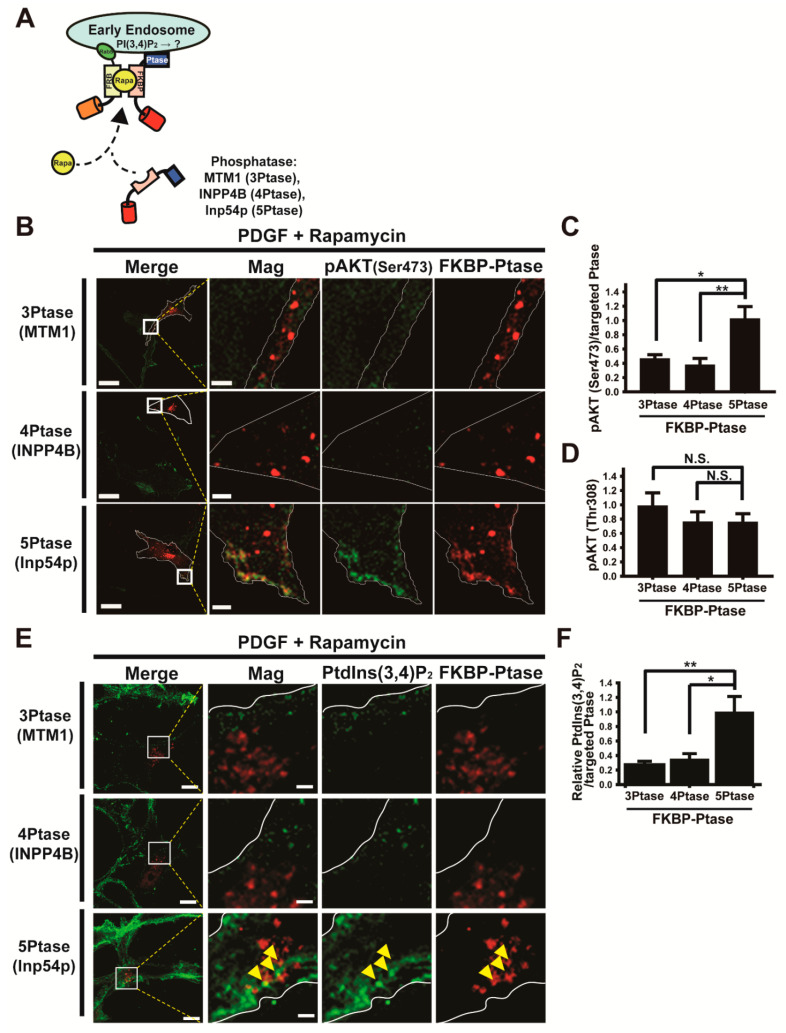
Activation of AKT by mTORC2 is mediated by early endosomal PtdIns(3,4)P_2_: (**A**) schematic representation of the heterodimerization of mCherry-FKBP-MTM1 (3-phosphatase), mCherry-FKBP-SJ2-Sac1 (4-phosphatase), or CFP- FKBP-Inp54p (5-phosphatase) to early endosomal membrane-anchored iRFP-FRB-Rab5, following addition of rapamycin; (**B**) representative confocal images of U87MG cells coexpressing iRFP-FRB-Rab5 and mCherry-FKBP-MTM1 (top, red), mCherry-FKBP-SJ2-4Ptase (middle, red), or CFP-FKBP-Inp54p (bottom, red) in the presence of rapamycin (40 nM) during PDGF activation for 5 min. Cells were fixed and stained with antibodies to pAKT(Ser473) (green). Scale bars, 20 μm; insets, 2 μm; (**C**,**D**) the relative fluorescence intensity ratios of pAKT(Ser473) to targeted phosphatase, respectively, were evaluated at the early endosomes in the presence of rapamycin (40 nM) during PDGF–BB activation for 5 min (**C**). The relative fluorescence intensity ratios of total pAKT(Thr308) to the targeted phosphatase were also measured in the presence of rapamycin (40 nM) during PDGF–BB activation for 5 min (**D**). Data are presented as the mean ± SEM from three independent experiments (*n* = 11–17 cells examined for each experiment), and *p*-values were calculated using Student’s two-tailed *t*-test. * *p* < 0.05, ** *p* < 0.01, N.S., not significant; (**E**) representative confocal images of U87MG cells coexpressing iRFP-FRB-Rab5 and mCherry-FKBP-MTM1 (top, red), mCherry-FKBP-INPP4B-4Ptase (middle, red), or CFP-FKBP-Inp54p (bottom, red) in the presence of rapamycin (40 nM) during PDGF–BB activation for 5 min. Cells were fixed and stained with antibodies to PtdIns(3,4)P_2_ (green). Arrowheads indicate PtdIns(3,4)P_2_ in targeted 5-phosphatase. Scale bars, 10 μm; insets, 1 μm; (**F**) the relative fluorescence intensity ratios of PtdIns(3,4)P_2_ to targeted phosphatase, respectively, were evaluated at the early endosomes in the presence of rapamycin (40 nM) during PDGF–BB activation for 5 min. Data are presented as the mean ± SEM from three independent experiments (*n* = 10–11 cells examined for each experiment), and *p*-values were calculated using Student’s two-tailed *t*-test. * *p* < 0.05, ** *p* < 0.01.

**Figure 6 cancers-13-02405-f006:**
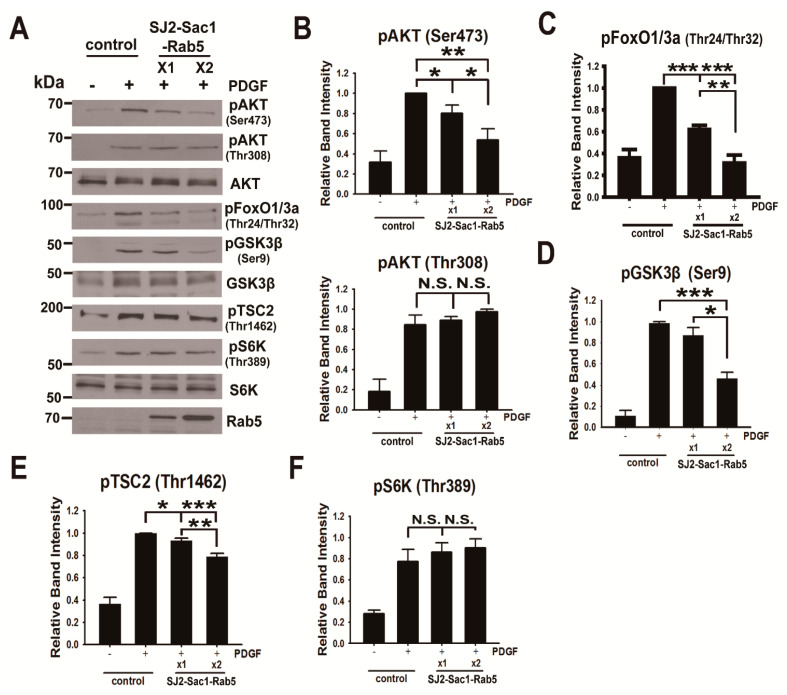
Expression of endosome-targeting SJ2-Sac1-Rab5 inhibits a subset of AKT signaling: (**A**) expression of phosphatidylinositol-4-phosphatase (synaptojanin2-Sac1 domain) targeted to the early endosome (SJ2-4Ptase-Rab5) diminished AKT phosphorylation at Ser473 in a concentration-dependent manner but not AKT phosphorylation at Thr308 in U87MG cells during PDGF–BB (50 ng/mL) stimulation for 5 min. Lysates from cells expressing control or SJ2-4Ptase-Rab5 were subjected to immunoblot analysis with antibodies to the indicated proteins; (**B**) the relative immunoblot signal intensity of pAKT(Ser473) and pAKT(Thr308) was determined as the mean ± SEM from three independent experiments, and *p*-values were calculated using Student’s two-tailed *t*-test. * *p* < 0.05, ** *p* < 0.01, N.S., not significant; (**C**–**F**) the relative immunoblot signal intensity of pFoxO1/3a(Thr24/Thr32) (**C**), pGSK3β(Ser9) (**D**), pTSC2(Thr1462) (**E**), and pS6K(Thr389) (**F**) was determined as the mean ± SEM from three independent experiments, and *p*-values were calculated using Student’s two-tailed *t*-test. * *p* < 0.05, ** *p* < 0.01, *** *p* < 0.001, N.S., not significant.

**Figure 7 cancers-13-02405-f007:**
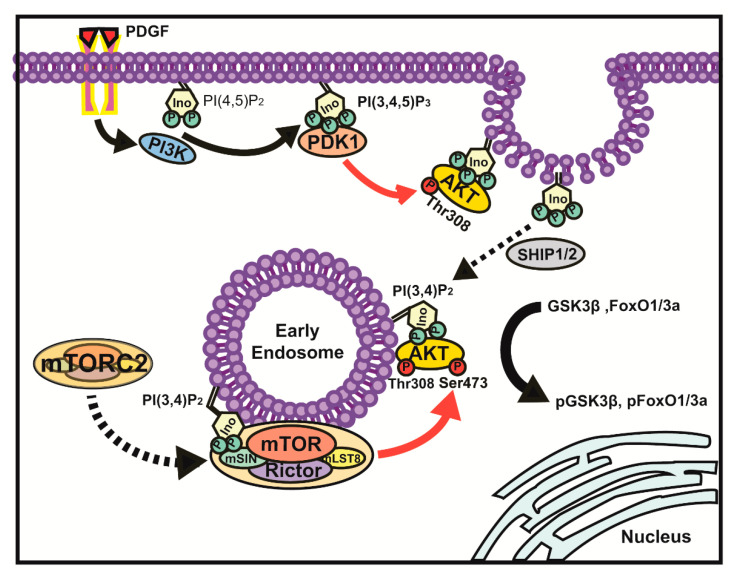
Proposed scheme for the mechanism of AKT activation by mTORC2 in early endosomes during growth factor stimulation. PDGF-dependent PI3K activation triggers AKT phosphorylation through PDK1 and mTORC2. PtdIns(3,4,5)P_3_ molecules produced by class I PI3K at the plasma membrane provide a binding platform for PDK1 and AKT through each protein’s PH domain. PtdIns(3,4)P_2_ molecules produced predominantly by SHIP1/2 5-phosphatase from PtdIns(3,4,5)P_3_ play a role in connecting mTORC2 to AKT by providing a binding site for mSIN, a component of the complex, and AKT. This local activation of AKT is required for a subset of substrates represented by GSK3β and FoxO1/3a.

## Data Availability

Data and information are included in the article or Appendix A or are available from the authors upon reasonable request.

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
