# Peer review of "Endosomal mTORC2 Is Required for Phosphoinositide-Dependent AKT Activation in Platelet-Derived Growth Factor-Stimulated Glioma Cells"

_cancers, 2021, doi:10.3390/cancers13102405_

Round 1

Reviewer 1 Report

Re: Manuscript ID: Cancers (ISSN 2072-6694)/cancers-1203793

Title: Endosomal mTORC2 Is Required for Phosphoinositide-Dependent AKT Activation in Platelet-Derived Growth Factor-Stimulated Glioma Cells

Authors: Suree Kim , Sukyeong Heo , Joseph Brzostowski , Dongmin Kang

In this present study, Kim et al.  investigated the role of serine/threonine kinase AKT during phosphatidylinositol 3-kinase (PI3K)-driven cell signal transduction in response to extracellular stimuli, especially the mechanism underlying target of rapamycin complex 2 (mTORC2) phosphorylation of AKT at Ser473 in the cellular endomembrane system. The authors demonstrated that endocytosis is required for AKT activation through phosphorylation at Ser473 via mTORC2 using platelet-derived growth factor-stimulated U87MG glioma cells. Furthermore, this present study showed that endosomal phosphoinositide, such as PtdIns(3,4)P2, provides a binding platform for mTORC2 to phosphorylate AKT Ser473 in endosomes through mammalian Sty1/Spc1-interacting protein (mSIN). The authors concluded that early endosomal events induced by endocytosis, together with the previously identified AKT activation by PtdIns(3,4,5)P3, contribute to strength-ening of the transduction of AKT signaling through phosphoinositide.

The results reported represent a notable advance in search of the significance of mTORC2 pathway in glioma progress. In my opinion, the manuscript is suitable for publication in Cancers, after the authors have addressed my comments and questions.

  1. In general, I found the experiments are technically sound and scientifically valid. Detailed and systematic experimental studies are presented. But the manuscript is not well organized. There are too much unrelated information that makes the manuscript hard to read for even the researchers in this field. If the authors want to make it a productive exercise, they need to have a clear idea of which kind of information they need to get in the first place, and then focus on that aspect. I strongly recommend the authors trim off about 40% of the text in the results, move most of the methods and results parts into supplement materials, so there is an easy flow of the description of the study. In that way, I assume the reference list will be shortened to about half.
  2. Even though there are important advances in the design, the analysis was repeating what has been done in many previous papers. Given this fact, the analysis was also far too lengthy for a compact journal, as if it was carrying an aspect of novelty. The discussion should be organized more systemically with more clear statement.
  3. U87MG is a cell line developed 50 years ago. There are reports that the DNA profile of the current cell line differs from that of the original cells with unknown origins. This misidentification of a widely studied cell line reinforces the need for researchers to carefully validate the cell lines used in their research. However, this study was solely based on this cell line. Using other cell lines, and recently cultured primary brain cancer cells to validate the findings is highly appreciated.
  4. There are many typos and writing inconsistencies across the paper, a through proofreading is needed. For example, in the methods part, the authors stated that “Mycoplasma testing was conducted periodically on both cell lines. Cell lines were subjected to transient transfection with expression plasmids using the Effectene reagent (QIAGEN, Germany) or the Neon transfection method (Thermo Fisher Scientific, USA), according to the manufacturers’ manual.”, however, I only found U87MG was used.

Author Response

Dear Editor

Thank you for your letter of April 22, 2021, allowing us to submit a revised version of our manuscript (ID: cancers-1203793) entitled “Endosomal mTORC2 Is Required for Phosphoinositide-Dependent AKT”. We are grateful for your explicit instructions for revision of our manuscript. We have revised the manuscript as suggested by the reviewers and incorporated our responses to the reviewers’ comments. Our responses and subsequent modifications are described below. The responses are placed below specific comments/questions (shown in italics) taken from the reviewers. We would like to thank reviewers for valuable comments.

Reviewer #1

In this present study, Kim et al. investigated the role of serine/threonine kinase AKT during phosphatidylinositol 3-kinase (PI3K)-driven cell signal transduction in response to extracellular stimuli, especially the mechanism underlying target of rapamycin complex 2 (mTORC2) phosphorylation of AKT at Ser473 in the cellular endomembrane system. The authors demonstrated that endocytosis is required for AKT activation through phosphorylation at Ser473 via mTORC2 using platelet-derived growth factor-stimulated U87MG glioma cells. Furthermore, this present study showed that endosomal phosphoinositide, such as PtdIns(3,4)P2, provides a binding platform for mTORC2 to phosphorylate AKT Ser473 in endosomes through mammalian Sty1/Spc1-interacting protein (mSIN). The authors concluded that early endosomal events induced by endocytosis, together with the previously identified AKT activation by PtdIns(3,4,5)P3, contribute to strength-ening of the transduction of AKT signaling through phosphoinositide.

The results reported represent a notable advance in search of the significance of mTORC2 pathway in glioma progress. In my opinion, the manuscript is suitable for publication in Cancers, after the authors have addressed my comments and questions.

  1. In general, I found the experiments are technically sound and scientifically valid. Detailed and systematic experimental studies are presented. But the manuscript is not well organized. There are too much unrelated information that makes the manuscript hard to read for even the researchers in this field. If the authors want to make it a productive exercise, they need to have a clear idea of which kind of information they need to get in the first place, and then focus on that aspect. I strongly recommend the authors trim off about 40% of the text in the results, move most of the methods and results parts into supplement materials, so there is an easy flow of the description of the study. In that way, I assume the reference list will be shortened to about half.

To trim off the text in the results, we have moved previous Figure 7 (AKT signaling via Ser473 phosphorylation is regulated by mTORC2 in PDGF-stimulated cells) and the description of Figure 7 into supplementary material.

  1. Even though there are important advances in the design, the analysis was repeating what has been done in many previous papers. Given this fact, the analysis was also far too lengthy for a compact journal, as if it was carrying an aspect of novelty. The discussion should be organized more systemically with more clear statement.

We have moved previous Figure 7 into supplementary material. The discussion has been reorganized and improved with more clear statements, as suggested by reviewer #1.

  1. U87MG is a cell line developed 50 years ago. There are reports that the DNA profile of the current cell line differs from that of the original cells with unknown origins. This misidentification of a widely studied cell line reinforces the need for researchers to carefully validate the cell lines used in their research. However, this study was solely based on this cell line. Using other cell lines, and recently cultured primary brain cancer cells to validate the findings is highly appreciated.

We provide the result from another mammalian cell line, epidermal growth factor (EGF)-stimulated A431 epidermoid carcinoma cells, below. The result demonstrates the presence of mTORC2 in early endosomes containing PtdIns(3,4)P2 during stimulation with EGF in A431 cells. Due to limited space in the text, the result is shown in this response letter only.

Figure. Localization of mSIN, a component of mTORC2, and PtdIns(3,4)P2 in Rab5-containing endosomes of EGF-stimulated A431 cells. Confocal microscopy of A431 cells expressing RFP-mSIN (red) activated with EGF (200 ng/mL) for 3 min and stained with antibodies specific to PtdIns(3,4)P2 (green) and Rab5 (blue). The areas indicated by boxes are shown at higher magnification in the insets. Scale bars, 10 μm; insets, 1 μm.

  1. There are many typos and writing inconsistencies across the paper, a through proofreading is needed. For example, in the methods part, the authors stated that “Mycoplasma testing was conducted periodically on both cell lines. Cell lines were subjected to transient transfection with expression plasmids using the Effectene reagent (QIAGEN, Germany) or the Neon transfection method (Thermo Fisher Scientific, USA), according to the manufacturers’ manual.”, however, I only found U87MG was used.

We have corrected typographical errors and inconsistencies through proofreading.

Reviewer 2 Report

While EGF-induced receptor activated is well studied, PDGFR activation is less so. It is mainly assumed that most RTKs follow a similar pattern of activation. In this manuscript, the authors look at PDGFR stimulation and endocytosis upon ligand binding. They present data suggesting the role mTORC2 localisation to early endosome and AKT activation. Overall, the study is well designed and data of high quality and includes some interesting methodologies. Specific comments are as follows:

Figure 2C&D: Do the GFP-mSIN-RBD-PH spots correspond to early endosomes? A colocalisation experiment is missing here to confirm their identity.

Figure 5: can the authors confirm that the rapamycin-induced localization of these phosphatases to endosomes results in reduced lipid species? This could be done using the lipid probes/antibodies in pervious experiments.

Discussion: as the authors focus their studies on PDGF-induced AKT activation, they should highlight this in the discussion.

Minor:

The domains included in the following constructs GFP-mSIN-RBD-PH & GST-EEA1-Rab5BD should be better described in the text.

Supplementary movies are difficult to follow. Perhaps an inclusion of arrows to monitor specific puncta and snapshots of puncta would be helpful to highlight the intended message.

Line 463-464: “we next investigated whether mSIN can bind to PtdIns(3,4)P2 in early endosomes of”, binding between PIP2 and mSIN cannot be concluded from the imaging assays. Similarly, statement on line 471-472. These need to be adjusted. Alternatively, the authors should conduct lipid-protein binding assays.

The authors should clarify throughout the text that they are using PDGF-BB to stimulate cells and scFv-Cκ-scFv to look at PDGFRB (instead of investigating PDGFRA activity). 

Author Response

Dear Editor

Thank you for your letter of April 22, 2021, allowing us to submit a revised version of our manuscript (ID: cancers-1203793) entitled “Endosomal mTORC2 Is Required for Phosphoinositide-Dependent AKT”. We are grateful for your explicit instructions for revision of our manuscript. We have revised the manuscript as suggested by the reviewers and incorporated our responses to the reviewers’ comments. Our responses and subsequent modifications are described below. The responses are placed below specific comments/questions (shown in italics) taken from the reviewers. We would like to thank reviewers for valuable comments.

Reviewer2

While EGF-induced receptor activated is well studied, PDGFR activation is less so. It is mainly assumed that most RTKs follow a similar pattern of activation. In this manuscript, the authors look at PDGFR stimulation and endocytosis upon ligand binding. They present data suggesting the role mTORC2 localisation to early endosome and AKT activation. Overall, the study is well designed and data of high quality and includes some interesting methodologies. Specific comments are as follows:

Figure 2C&D: Do the GFP-mSIN-RBD-PH spots correspond to early endosomes? A colocalisation experiment is missing here to confirm their identity.

We have added the confocal microscopy image showing that GFP-mSIN-RBD-PH spots correspond to early endosomes containing constitutively active GTPase in Figure 2E.

Figure 5: can the authors confirm that the rapamycin-induced localization of these phosphatases to endosomes results in reduced lipid species? This could be done using the lipid probes/antibodies in pervious experiments.

We provide the result that U87MG cells expressing mCherry-INPP4B show reduced PtdIns(3,4)P2 level compared with cells expressing mCherry below. mCherry-INPP4B activity was verified with PtdIns(3,4)P2 as a substrate. mCherry-FKBP-MTM1 activity and CFP-FKBP-Inp54p activity have been well verified in previous works (for FKBP-MTM1: Hammond GRV et al., 2014, Journal of Cell Biology, 205(1): 113-126; Uchida Y et al., 2017, Molecular Pharmacology, 91:65-73; for FKBP-Inp54p: Suh BC et al., 2006, Science 314: 1454-1457; Kim SJ et al., 2017 Scientific Reports 7: 3351). FKBP-INPP4B has been verified in a previous work (Goulden BD et al., 2019, Journal of Cell Biology, 218(3): 1066-1079). Due to limited space in the text, the result is shown in this response letter only.

Figure. mCherry-INPP4B expression decreases PtdIns(3,4)P2 level. (A) Immunofluorescence analysis of PtdIns(3,4)P2 (green) intensity in U87MG cells expressing mCherry or mCherry-INPP4B were treated with PDGF-BB (50 ng/ml) for 15 min. Nuclei were stained with DAPI (blue) in the merged image. (B) Quantification of different relative PtdIns(3,4)P2 levels from (A). Data are means ± S.E.M. of three independent experiments (n = 15-18 areas for each condition in each experiment). The average pixel intensity of 3 areas per cell edge was measured. **P < 0.01.

Discussion: as the authors focus their studies on PDGF-induced AKT activation, they should highlight this in the discussion.

We have highlighted our study on PDGF-induced AKT activation in the discussion with more clear statements.

Minor:

The domains included in the following constructs GFP-mSIN-RBD-PH & GST-EEA1-Rab5BD should be better described in the text.

We have added the domains of GFP-mSIN-RBD-PH and GST-EEA1-Rab5BD in the text (Results and Materials & Methods).

Supplementary movies are difficult to follow. Perhaps an inclusion of arrows to monitor specific puncta and snapshots of puncta would be helpful to highlight the intended message.

We have added puncta arrows in supplementary movies.

Line 463-464: “we next investigated whether mSIN can bind to PtdIns(3,4)P2 in early endosomes of”, binding between PIP2 and mSIN cannot be concluded from the imaging assays. Similarly, statement on line 471-472. These need to be adjusted. Alternatively, the authors should conduct lipid-protein binding assays.

We have adjusted and improved the description.

Line 474-476: “we next investigated whether mSIN can colocalize to the early endosomes containing PtdIns(3,4)P2 in PDGF-activated cells using an antibody against PtdIns(3,4)P2.”

Line 483-485: “Structured illumination microscopy (SIM) revealed that mSIN colocalized with PtdIns(3,4)P2 at a distance of 30–60 nm. There are also regions of mSIN that do not exhibit PtdIns(3,4)P2 (Figure 4E).”

The authors should clarify throughout the text that they are using PDGF-BB to stimulate cells and scFv-Cκ-scFv to look at PDGFRB (instead of investigating PDGFRA activity).

We have clarified the use of PDGF-BB throughout the text.

Round 2

Reviewer 1 Report

The authors have addressed satisfactorily the points I raised previously, in light of the response to both reviewers.  With the additional data presented and discussion, the paper has substantially improved.

Author Response

Dear Editor

Thank you for your letter of May 8, 2021, allowing us to submit a revised version of our manuscript (ID: cancers-1203793) entitled “Endosomal mTORC2 Is Required for Phosphoinositide-Dependent AKT Activation in Platelet-Derived Growth Factor-Stimulated Glioma Cells”. We have revised the manuscript as suggested by the reviewers. We have changed the size of figures to reduce empty space in the manuscript. New changes were shown in red in the manuscript. Our responses and subsequent modifications are described below. The responses are placed below specific comments/questions (shown in italics) taken from the reviewers. We very much appreciate reviewers for valuable comments.

Reviewer 1

The authors have addressed satisfactorily the points I raised previously, in light of the response to both reviewers. With the additional data presented and discussion, the paper has substantially improved.

Thanks. We have carefully double-checked and revised the manuscript for English language and style.

Reviewer 2 Report

The authors have addressed all my comments.

The one remaining point to address is the text related to figure 5, following on from my previous comment. Perhaps the authors could clarify the specific phosphoinositide species that can be targetted by the phosphotases. This can be added to figure 5A. In addition, I understand the authors' concern about word limitation, but I think the references they cite in their rebuttal that show that the FKBP system works for the phosphatases assayed should be included in the manuscript (if they haven't been already).

Author Response

Dear Editor

Thank you for your letter of May 8, 2021, allowing us to submit a revised version of our manuscript (ID: cancers-1203793) entitled “Endosomal mTORC2 Is Required for Phosphoinositide-Dependent AKT Activation in Platelet-Derived Growth Factor-Stimulated Glioma Cells”. We have revised the manuscript as suggested by the reviewers. We have changed the size of figures to reduce empty space in the manuscript. New changes were shown in red in the manuscript. Our responses and subsequent modifications are described below. The responses are placed below specific comments/questions (shown in italics) taken from the reviewers. We very much appreciate reviewers for valuable comments.

Reviewer 2

The one remaining point to address is the text related to figure 5, following on from my previous comment. Perhaps the authors could clarify the specific phosphoinositide species that can be targetted by the phosphotases. This can be added to figure 5A. In addition, I understand the authors' concern about word limitation, but I think the references they cite in their rebuttal that show that the FKBP system works for the phosphatases assayed should be included in the manuscript (if they haven't been already).

We have added new data demonstrating the rapamycin-induced localization of 3-phosphatase or 4-phosphatase to early endosomes results in reduced endosomal PtdIns(3,4)P2 but that of 5-phosphatase dose not (Figure 5E,F) in the manuscript. Description of the result has been added in Line 531-535 and Line 557-564 (in red).

The references for FKBP-MTM1, FKBP-INPP4B, and FKBP-Inp54p activity were included in the manuscript as suggested by reviewer #2. We revised the manuscript for English language and style.